

# Harnessing citizen science for marine conservation in Malta: a comparative analysis of GAM and MaxEnt models in bottlenose dolphin habitat mapping

Francesca Soster[1,*], Tim Awbery[2,*], Nina Vérité–Taulet[3], Timothy Zammit[3] and Kimberly Terribile[4]

[1] Applied Research and Innovation Centre, Malta College of Arts Science & Technology (MCAST), Paola, Malta
[2] Marine Mammal Research Team, Scottish Association for Marine Science, Oban, United Kingdom
[3] Discover the Blu, San Pawl il-Baħar, Malta
[4] Centre for Agriculture, Aquatics and Animal Sciences, Malta College of Arts, Science and Technology (MCAST), Paola, Malta
[*] These authors contributed equally to this work.

Corresponding author
Francesca Soster,
francesca.soster@mcast.edu.mt

## ABSTRACT

**Background**. Species distribution models (SDMs) are powerful tools for informing conservation, particularly for highly mobile marine species such as common bottlenose dolphins (*Tursiops truncatus*). In Maltese waters, the limited availability of data on this species has constrained the effectiveness of conservation efforts. Despite the designation of offshore Special Areas of Conservation (SACs), key coastal regions need more detailed spatial studies to support evidence-based management.

**Methods**. In this study, we analyzed and compared the outputs of a generalized additive model (GAM) and a maximum entropy (MaxEnt) model to assess summer habitat suitability for bottlenose dolphins within a coastal SAC in Malta. The models were informed by presence-only data collected through systematic surveys and a citizen science campaign, integrated with environmental and anthropogenic predictors including chlorophyll-a concentration, sea surface temperature anomaly, slope, and distance to aquaculture sites.

**Results**. Both modeling approaches identified high habitat suitability in shallow, nearshore regions, with chlorophyll-a concentration and proximity to aquaculture sites emerging as the most important predictors. Slope and sea surface temperature anomaly contributed less substantially. The two models showed spatial agreement in highlighting these nearshore areas as core habitats, though GAM predicted a broader extent of suitable habitat, whereas MaxEnt results were more spatially restricted. Both models demonstrated strong predictive performance (AUC > 0.85), reinforcing the ecological relevance of the identified drivers.

**Conclusion**. This study demonstrates the potential of integrating opportunistic data with SDMs to support habitat assessments in data-limited contexts. The use of complementary modeling approaches provides robust insights into species–environment relationships. These results aim to guide spatial planning and future assessments of conservation priorities in Maltese coastal waters.

# INTRODUCTION

Understanding species distribution and habitat use is fundamental for developing effective conservation strategies (*Rodríguez et al., 2007*; *Guisan et al., 2013*). However, assessing these aspects can be particularly challenging for highly mobile species like cetaceans (*Fernandez, Sillero & Yesson, 2022*). In this context, species distribution models (SDMs) offer a solid foundation for gaining valuable ecological insights, particularly in data-deficient regions (*Redfern et al., 2006*; *Fiedler et al., 2018*; *Fernandez et al., 2021*). By evaluating the relationships between cetacean populations and the environment, as well as anthropogenic factors, SDMs contribute to a more comprehensive understanding of species ecology and potential conservation priorities (*Rodríguez et al., 2007*; *Guisan et al., 2013*; *Marshall, Glegg & Howell, 2014*; *Pace, Tizzi & Mussi, 2015*; *Giralt Paradell, Díaz López & Methion, 2019*).

The ability of SDMs to assess species distributional ranges *via* predictive modeling (*Anderson, Lew & Peterson, 2003*; *Pitchford et al., 2016*) or to identify the environmental drivers of distribution through explanation modeling (*Azzellino et al., 2008*; *La Manna et al., 2023b*) has led to an increase in their use in recent years for cetacean species (*Pasanisi et al., 2024*). When used together, predictive and explanation modeling provides information on the distribution of habitats that are suitable for species survival (*Hirzel & Le Lay, 2008*). Within this framework, habitat suitability models serve as practical tools to estimate the ecological conditions and areas within which a species is most likely to occur (*Guisan & Thuiller, 2005*).

Among the most widely adopted modeling techniques are machine learning algorithms such as maximum entropy (MaxEnt) models (*Melo-Merino, Reyes-Bonilla & Lira-Noriega, 2020*), which estimate the habitat suitability from presence-only data using the principle of maximum entropy (*Phillips, Anderson & Schapire, 2006*), and statistical approaches such as generalized additive models (GAMs), which model non-linear relationships between species occurrence (*e.g.*, presence/absence) and environmental factors (*Hastie & Tibshirani, 1986*). Both methods enable researchers to identify critical conservation areas and mitigate potential threats by analyzing species' habitat preferences (*Rodríguez et al., 2007*; *Guisan et al., 2013*). Furthermore, they help predict how marine species respond to environmental changes, contributing to the development of targeted conservation strategies (*Giralt Paradell, Díaz López & Methion, 2019*; *Díaz López & Methion, 2024*).

The performance and reliability of these models are inherently dependent on several factors, including model parametrization (*Elith et al., 2006*), the ecological relevance and statistical independence of environmental predictors (*Guisan & Zimmermann, 2000*; *Dormann et al., 2013*), the spatial and temporal scale at which models are implemented (*Elith & Leathwick, 2009*) and the overall quality, completeness, and bias of species occurrence records (*Hernandez et al., 2006*; *Araújo & Guisan, 2006*; *Merow, Smith & Silander, 2013*).

Dedicated surveys, although considered robust, are often resource-intensive and limited by logistical and financial constraints, resulting in incomplete spatial coverage and potential underrepresentation of species distribution (*Evans & Hammond, 2004*; *Meyer et al., 2015*). Recent studies have demonstrated that non-traditional data sources, such as opportunistic and citizen science data (*Giovos et al., 2016*; *Pace et al., 2019*; *Robbins, Babey & Embling, 2020*), offer a cost-effective approach to addressing specific research challenges, facilitating the generation of estimates of cetacean distribution (*Fernandez et al., 2021*). In the study of species inhabiting dynamic environments, non-traditional data sources offer several advantages over conventional survey methodologies. For instance, they can enhance spatial and temporal coverage and reduce logistical constraints, as they often do not require specialized equipment or dedicated field effort (*Robbins, Babey & Embling, 2020*; *Corr et al., 2024*). Nevertheless, the inherent biases and variability associated with opportunistic data necessitate careful quality assessment and data integration techniques to ensure their effective use in large-scale modeling efforts (*Isaac et al., 2020*; *Martino et al., 2021*).

Given these challenges, the integration of SDMs with non-traditional data sources presents a promising approach for studying cetaceans in regions where systematic monitoring is limited. In Maltese waters, the common bottlenose dolphin (*Tursiops truncatus*) represents one of the most frequently observed cetacean species (*Notarbartolo Di Sciara, 2002*) and plays an important ecological role as a top predator within the marine ecosystem. Bottlenose dolphins are protected under a framework of national, regional and international legal frameworks. At the national level, all cetacean species are protected through the Flora, Fauna and Natural Habitats Protection Regulations (S.L. 549.44), which transpose the European Union (EU) Habitats Directive 92/43/EEC into Maltese law. In particular, the bottlenose dolphin is listed in Annex II of the Directive, which requires the designation of Special Areas of Conservation (SACs) forming part of the Natura 2000 network, and in Annex IV as a species of community interest in need of strict protection.

Despite its protected status, the limited availability of systematic data on bottlenose dolphins in Maltese waters presents challenges for fully supporting conservation planning. Previous studies in the region have suggested a coastal distribution and spatial association between dolphin presence and anthropogenic features such as aquaculture sites, which may influence foraging behavior and local prey availability (*Laspina, Terribile & Said, 2022*; *Soster et al., 2025*). Recent modeling efforts have highlighted potential gaps in the spatial alignment between currently designated Natura 2000 sites and areas predicted to be of high habitat suitability for bottlenose dolphins around the Maltese archipelago (*Soster et al., 2025*). Specifically, although SACs have been designated for bottlenose dolphin conservation in Malta, these mainly encompass offshore regions, while key coastal regions remain outside protected boundaries. This highlights the need for further in-depth investigation of bottlenose dolphin spatial ecology and habitat use in these areas, particularly through temporally and spatially specific studies that can build upon existing baseline knowledge. Moreover, dolphins in the region are exposed to a range of human pressures, including maritime traffic, aquaculture, and unregulated recreational activities (*Said et al., 2017*; *Filletti et al., 2023*; *Mizzi et al., 2024*; *Soster et al., 2025*). They are additionally subject to environmental pressures, such as marine heatwaves (*Garrabou et al., 2022*), which require

further investigation to be adequately addressed under existing management frameworks. These conservation gaps highlight the urgent need for spatially explicit data on dolphin distribution and habitat use to inform evidence-based conservation strategies in Malta's dynamic and heavily used coastal waters.

In light of these gaps, this case study aims to improve understanding of bottlenose dolphin habitat suitability using presence-only data from opportunistic research platforms and citizen science. We apply GAM and MaxEnt approaches within a heavily impacted coastal SAC that, despite its designation, does not explicitly include bottlenose dolphins as a target species. While this study is conservation-focused and not a methodological comparison of modeling techniques, the inclusion of both models allows for a more nuanced interpretation of species–habitat relationships and enhances stakeholder confidence in the results. By analyzing and comparing the outputs, we identify summer habitat suitability patterns and assess the influence of environmental and anthropogenic drivers. The ultimate long-term goal of this work is to support evidence-based spatial planning and inform the potential inclusion of bottlenose dolphins within the site's conservation framework.

## MATERIALS & METHODS

### Study area

The study area covers 159 km² and is situated on the northeastern side of the Maltese archipelago, extending from central Malta to the northern coast of Gozo, with a maximum depth of 100 m (Fig. 1). The area includes the Natura 2000 site Żona fil-Baħar bejn Il-Ponta ta ' San Dimitri (Għawdex)u Il-Qaliet, designated as a SAC for the protection of the Maltese top-shell (*Steromphala nivosa*) and the loggerhead turtle (*Caretta caretta*). Although bottlenose dolphins are known to occur in the area, they are not currently listed among the protected species for this site (*EUNIS, 2025*). This coastal SAC hosts a wide range of important habitats, such as *Posidonia oceanica* meadows, maërl beds, and *Cymodocea nodosa* meadows. However, this site is affected by numerous human activities, including four main finfish aquaculture sites and one bunkering area for large vessels (Fig. 1). Furthermore, pleasure boating is especially prevalent, driven by the high concentration of marinas and the region's appeal as a tourist destination.

### Occurrence data

Occurrence data of bottlenose dolphins were collected using two complementary approaches: (i) dedicated boat surveys were conducted between July and September 2024 aboard a six-meter rigid-hull boat equipped with a 225 hp engine, following a standardized observational protocol during whale watching activities, permitted by the Environment and Resources Authority (EP 0249/24; Supplementary Material 1A); (ii) opportunistic data were gathered through a citizen science campaign, which encouraged sea users to report dolphin sightings by submitting information on coordinates, group size and composition, behavior, and photographs (Supplementary Material 1B). Outreach included the distribution of flyers to marinas, diving centers, charter companies, and tour operators across the study area. In addition, opportunistic reports shared *via* social

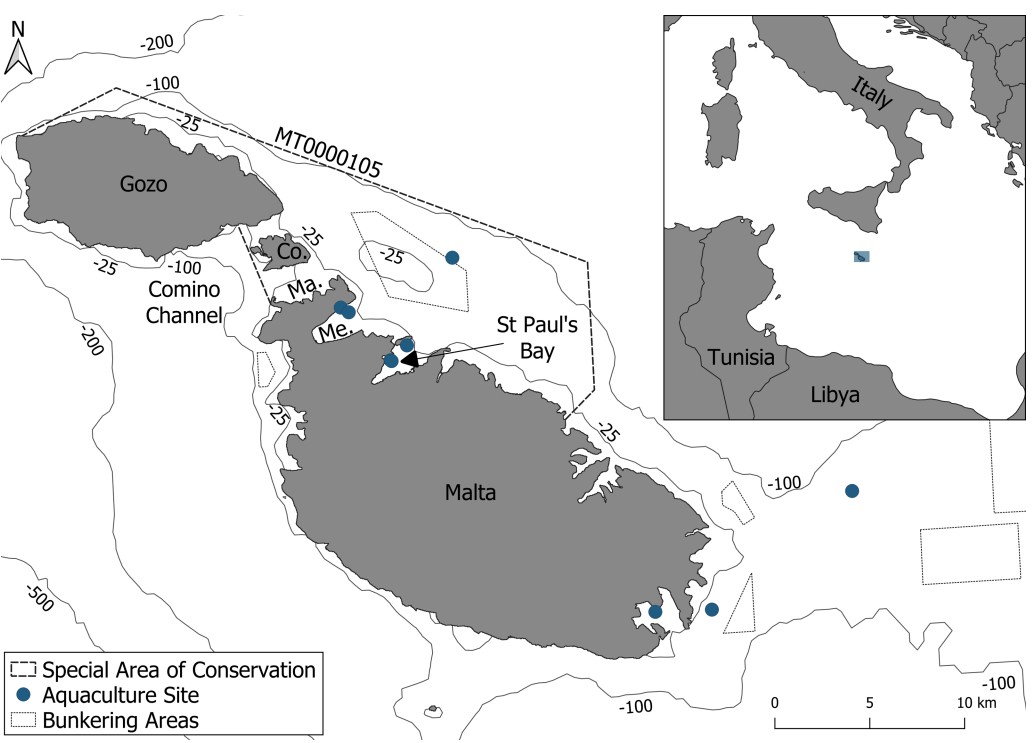

**Figure 1** **The survey area consisting of the Special Area of Conservation (MT0000105).** Within the Special Area of Conservation (SAC), delineated by a dashed line, are aquaculture sites and a bunkering area delineated by blue points and a dotted black line respectively. Abbreviations: Co., Comino; Me., Mellieha Bay; Ma., Marfa Bay.

media platforms were also collected and screened (*Pace et al., 2019*). All records were validated through expert verification before the inclusion in the database, a method widely utilized for its high level of accuracy (*Yu, Wong & Hutchinson, 2010*; *Baker et al., 2021*). The validation process considered the reliability of the observer, the availability of supporting visual evidence (*e.g.*, videos or photographs), and existing knowledge of the local bottlenose dolphin population (*Bonter & Cooper, 2012*).

Data collection was limited to the summer season due to resource availability and its alignment with the regional peak in dolphin sightings and more favorable sea conditions, thereby enhancing both detection probability and data quality. An effort-based GAM was initially generated, but the low number of presence points with associated effort meant that the results were not reliable (*i.e.,* deviance explained >99.9%). Thus, presence-only data were considered for two modeling approaches using a binomial GAM and a MaxEnt model.

## Environmental data

A number of static environmental variables were included in the models, including depth, slope, aspect, and distance from shore. Dynamic variables comprised sea surface temperature (SST), chlorophyll-a concentration, mixed layer depth, and salinity. The selection of these variables was informed by previous studies on bottlenose dolphins

(*Pitchford et al., 2016*; *Carlucci et al., 2016*; *La Manna, Ronchetti & Sarà, 2016*; *Fontanesi et al., 2024*). In addition, SST anomaly, chlorophyll-a anomaly, and distance from aquaculture sites were incorporated to account for the potential influence of direct and indirect anthropogenic impacts on habitat suitability (*Maricato et al., 2022*). Details of the downloaded datasets, spatial resolutions and sources are given in Table 1. All dynamic variables were downloaded at a daily temporal scale, mean averaged across the survey period and resampled to the resolution of the depth layer (Fig. S1). A number of the variables had a high level of correlation (Pearson's correlation coefficient $|r|>0.7$) and were removed from the analysis (*Dormann et al., 2013*). In order to decide which of a pair of correlated variables was removed, two single-variable GAMs were run, and the variable in the model with the lowest AIC value was selected. Additionally, following the fitting of the best-performing GAM, a check of concurvity (a non-linear equivalent of a collinearity test) was made. This diagnostic is specific to models like GAMs that include smooth functions and allow assessment of non-linear dependencies among predictors. This step is not applicable to MaxEnt models, as they do not estimate smooth functions and lack a direct analogue to concurvity diagnostics. The final set of environmental predictors retained after this process was used consistently in both the GAM and MaxEnt models to ensure comparability of outputs.

## Pseudoabsences and background points selection

Whilst both GAM and MaxEnt are used regularly in species distribution modeling, they differ notably in how they handle absence information and the selection of pseudo-absence or background points. GAMs require both presence and absence (or pseudo-absence) data, and model performance can be influenced by the ratio between these points. A common recommendation is to use a presence-to-pseudo-absence ratio between 1:1 and 1:5, although using a large number of pseudo-absences (*e.g.*, 10,000) with equal weighting can also enhance model accuracy, particularly in large or heterogeneous study areas (*Barbet-Massin et al., 2012*). In contrast, MaxEnt operates on presence-only data and uses background points to represent the available environmental space, without assuming true absences. The default setting in MaxEnt is 10,000 background points, which is generally suitable for most applications, though this number can be adjusted depending on the extent and resolution of the study area (*Phillips & Dudík, 2008*). These methodological differences influence model outputs and will be considered when comparing results.

To account for sampling bias and improve model accuracy, a bias raster was generated using a kernel density estimation based on the spatial distribution of dolphin sightings (Fig. S2). A focal (neighborhood-based) smoothing operation was applied to create a continuous surface representing relative sampling effort, constrained within a two-kilometer buffer around both dolphin sightings and the boundaries of the SAC. The resulting raster was normalized and used to guide the selection of pseudo-absence/background points. A total of 30,000 points were initially generated (Fig. S3) using the randomPoints function in the dismo package in R, version 1.3-16 (*Hijmans et al., 2024*).

**Table 1  Environmental and anthropogenic predictors used in the models, including data source and resolution.**

| Predictor | Details and description |
| --- | --- |
| Depth | Water depth was obtained from the GEBCO 2023 global bathymetric grid (15 arc-seconds, approximately 460 ×380 m at Malta's latitude spatial resolution) (https://www.gebco.net/data_and_products/gridded_bathymetry_data/). The bathymetry raster was clipped to the study area extent and projected to UTM Zone 33N to ensure accurate distance calculations. |
| Slope and Aspect | Slope and aspect were derived from the GEBCO bathymetry raster using the R package *terra*. Slope represents the rate of change in elevation, while aspect indicates the direction of the steepest slope. These rasters were generated at the same resolution as the input bathymetry grid (approximately 460 ×380 m). |
| Distance from Shore and Distance from Aquaculture Sites | Euclidean distance from each raster cell to the nearest coastline or aquaculture facility, calculated using the "Distance" tool in the *terra* package in R. The coastline was obtained from official Maltese marine boundaries, and aquaculture sites were manually digitised from government sources (https://msdi.data.gov.mt/geoportal.html). |
| Sea Surface Temperature (SST) | Daily mean SST values were obtained from the STREAM App (https://stream-srf.com/products/), providing high-resolution satellite-derived SST data (∼1 km grid resolution) based on Copernicus Marine Service inputs. SST values were averaged over the study period to generate a composite summer surface. |
| Chlorophyll-a | Daily chlorophyll-a concentration data were sourced from the STREAM App. The product integrates Copernicus Ocean Colour multi-sensor datasets at ∼1 km spatial resolution, using the MedOC4 algorithm for offshore waters and AD4 for coastal waters. Values were averaged over the summer period. |
| Mixed Layer Depth | Mixed layer depth was obtained from the Mediterranean Sea Physics Analysis and Forecast product (Copernicus Marine Service: https://marine.copernicus.eu/). The product provides daily gridded data at 1/24° (∼4 km) spatial resolution. |
| Salinity | Daily surface salinity data were also extracted from the Mediterranean Sea Physics Analysis and Forecast product (Copernicus Marine Service, 1/24° spatial resolution). Mean salinity values were calculated for the summer study period. |
| SST Anomaly | SST anomaly, representing the deviation of daily SST from the 30-year climatological mean (1989–2019), was obtained from the STREAM App. This allows identification of thermal anomalies across the study area at ∼1 km resolution. |
| Chlorophyll-a Anomaly | Chlorophyll-a anomaly data were sourced from the STREAM App, calculated as the difference between daily chlorophyll-a concentrations and the 26-year climatological mean (Sept 1997–Aug 2023). The product uses multi-sensor satellite observations and gap-free data at ∼1 km resolution. |

To estimate the optimal number of background points for GAMs, 30 values were randomly sampled from the generated dataframe using base R (v4.4.3; *R Core Team, 2025*). Presence and background data were split into 90% training and 10% testing sets. GAMs were fitted to the training data and evaluated on the test set using the area under the curve (AUC) and the true skill statistic (TSS). AUC is threshold-independent and widely used for its simplicity and interpretability (*Liu, White & Newell, 2011*), while TSS provides a threshold-dependent metric less sensitive to prevalence (*Shabani, Kumar & Ahmadi, 2018*). This process was repeated 50 times using k-values (wiggliness parameter) of 3, 5,

and 10, and background point values ranging from 30 to 250, exceeding the commonly recommended 1:1 to 1:5 presence-to-pseudo-absence ratio (*Barbet-Massin et al., 2012*). For the GAM, AUC stabilized beyond 50 background points (Fig. S4A), while TSS peaked between 50 and 90 (Fig. S4B). Models with $k = 3$ or 5 consistently outperformed those with $k = 10$. A 1:1 presence-to-background ratio and $k = 5$ were selected for final modeling.

The same procedure was applied to MaxEnt, with background values from 1,000 to 12,000 and beta (regularization multiplier) values from 1 to 5. For MaxEnt, AUC and TSS remained stable across most background values, with AUC between 0.87 and 0.89, and TSS between 0.50 and 0.55 (Figs. S5A, S5B). Given the small size (159 km$^2$) and homogeneity of the study area, 8,000 background points were selected. A beta value of 1 was retained, following *Phillips & Dudík (2008)* for small samples, and to avoid over-regularization, which may reduce ecological interpretability (*Merow, Smith & Silander, 2013*).

## Model specifications and evaluation

The GAM was fitted using the binomial family with a logit link function in the mgcv package in R (version 1.9-1). Variable selection was enabled using shrinkage smoothers, which apply penalties to smooth terms that do not contribute meaningfully to the model, effectively shrinking them towards zero and allowing their influence to be removed.

The MaxEnt model was fitted using specific parameters implemented through the dismo package in R. Linear, quadratic, and hinge features were included, while product features were excluded, as they are only enabled when the number of presence points exceeds 80 (*Phillips & Dudík, 2008*). Threshold features were also omitted to minimize model complexity (*Elith et al., 2011*). The model was run using 500 iterations and 8,000 background points.

For both models, a 10-fold cross-validation procedure (90% train/10% test) was conducted to evaluate predictive performance as previously done when dataset sizes are constrained (*Fielding & Bell, 1997*; *Breiner et al., 2015*; *Deneu et al., 2021*). This approach was selected to provide a robust estimate of model accuracy while minimizing potential biases related to overfitting and data imbalance. The dataset was stratified into ten folds, ensuring a balanced distribution of presence and pseudo-absence observations for the GAM, and presence and background observations for the MaxEnt model. Each fold was used once as a testing set, while the remaining nine folds served as the training set. This process was repeated iteratively so that each observation contributed to both training and validation. For each iteration, a model was fitted to the environmental predictors, and predictions of species presence probability were generated for the testing set. Model performance was evaluated using receiver operating characteristic (ROC) curve analysis (*Fielding & Bell, 1997*; *Phillips, Anderson & Schapire, 2006*; *Elith et al., 2006*). The AUC was calculated for each fold as a measure of predictive accuracy, with higher AUC values indicating better model performance (*Phillips, Anderson & Schapire, 2006*; *Elith et al., 2006*; *Merow, Smith & Silander, 2013*).

Following model fitting, uncertainty was quantified for both GAM and MaxEnt predictions to assess the reliability of habitat suitability outputs. For GAMs, standard errors were extracted directly from the model's built-in prediction outputs (*Wood, 2017*).

For MaxEnt, a bootstrap resampling approach was used, and the resulting raster stack of 500 replicate predictions was analyzed (*Elith & Leathwick, 2009*). In both cases, a mean habitat suitability map was generated to represent the central tendency of model outputs (*Phillips, Anderson & Schapire, 2006*). To further quantify uncertainty, the standard error was calculated for each raster cell, reflecting the absolute variability in predicted values, while the coefficient of variation (CV) was computed to express relative uncertainty as a percentage of the mean prediction (*Chen, Dimitrov & Meyers, 2019*). These uncertainty metrics were used to identify areas of high prediction confidence and regions where model outputs were more variable. Given the two different methodologies of extracting the coefficient of variation, these are not directly comparable between the two modeling techniques but provide a useful understanding of spatial uncertainty around the model. Similarly, the output predictions of GAMs and MaxEnt are different; GAMs use logistic regression (binomial family with a logit link) to predict the probability of a presence whereas MaxEnt returns a predicted suitability (*Guisan, Edwards & Hastie, 2002*; *Phillips, Anderson & Schapire, 2006*). Whilst not directly comparable, both methods can be used to determine the area which is most suitable for the species of interest.

## Contribution and importance of variables

Both GAMs and MaxEnt offer visual tools to help interpret how environmental variables influence species presence. In GAMs, the shape of the partial response curves illustrates the effect of each predictor on the log-odds of species presence (*Guisan, Edwards & Hastie, 2002*). In MaxEnt, response curves indicate the species' relative preference across the range of each environmental variable (*Phillips, Anderson & Schapire, 2006*). Given the differences, the two should not be directly compared. They can however, both be useful in interpreting the effects of variables and thus are both given in the results.

Similarly, MaxEnt also provides two metrics of variable importance: percent contribution, based on the increase in model gain during training, and permutation importance, which measures the decrease in model performance when a variable's values are randomly permuted (*Phillips, Anderson & Schapire, 2006*; *Elith et al., 2011*). While GAMs do not include built-in measures of variable importance, a comparable estimate can be derived by examining deviance explained. This involves removing one predictor at a time and comparing the deviance of the reduced model to the full model (*Wood, 2017*). As with the visual response curves, these importance measures are not directly comparable across modeling approaches, but each helps to understand the role of environmental drivers and has been presented accordingly.

## Investigating the additional value of including citizen science data

To assess the influence of citizen science data on model performance, both the GAM and MaxEnt models were run without the inclusion of citizen science records. Model parameters were kept identical to those used in the full models: a 1:1 ratio of presence to background points and a k-value of five for the GAM, and 8,000 background points with a beta value of one for MaxEnt.

## RESULTS

### Occurrence

A total of 17 boat-based surveys covered 484.8 km of on-effort trackline, with a mean vessel speed of 9.7 kt (SE ± 0.18). The survey effort was unevenly distributed across the study area, with the majority of effort concentrated in the northern coastal waters of Malta (Fig. 2). A total of 20 bottlenose dolphin sightings were recorded during these surveys, with most sightings occurring in shallow to moderately deep waters (<50 m) to the north of Malta. The encounter rate for boat-based surveys was 4.12 groups and 19.8 individuals per 100 km surveyed. The citizen science campaign contributed an additional 49 sightings within the study area, out of a total of 52 validated records submitted during the campaign period. Together, the combined dataset resulted in 69 sightings (Fig. 2).

### Collinearity tests

The collinearity test resulted in the retention of slope, aspect, chlorophyll-a, SST anomaly, and distance to aquaculture. The subsequent concurvity assessment for the GAM led to the removal of aspect due to high non-linear dependency with other predictors. As a result, the final models were fitted using chlorophyll-a, SST anomaly, slope and distance to aquaculture.

### Habitat suitability analysis with GAM

The GAM showed strong predictive performance, with a mean AUC of 0.86 (SE ± 0.02) under 10-fold cross-validation, indicating a high level of discriminatory power (*Swets, 1988*). All four explanatory variables included in the model were found to have statistically significant relationships with bottlenose dolphin occurrence. In terms of variable importance, the proportion of deviance explained was highest for chlorophyll-a (0.79) and distance to aquaculture (0.65), suggesting these were the strongest drivers of dolphin habitat suitability (Table 2). Although slope and SST temperature anomaly were statistically significant, they contributed less to the overall deviance explained (0.20 and 0.16, respectively), indicating a weaker influence on habitat suitability patterns.

The model predicted an increased probability of occurrence with increasing values of slope and chlorophyll-a and a decreased probability of occurrence with increasing values of SST anomaly and distance to aquaculture facilities (Fig. 3).

The mean prediction map generated across 1,000 iterations showed that the highest habitat suitability for bottlenose dolphins was concentrated around Mellieha Bay and St Paul's Bay. Additional areas of moderate suitability were identified in the Comino Channel, in proximity to the bunkering area and along the north-eastern coast of Gozo (Fig. 4). The uncertainty associated with the spatial predictions, represented by the coefficient of variation across, was generally low within the borders of the SAC but higher at the outer edges of the survey region.

### Habitat suitability analysis with MaxEnt

The MaxEnt model demonstrated strong predictive performance (*Swets, 1988*), with a mean AUC of 0.89 (SE ± 0.0001) under 10-fold cross-validation. The regularized training
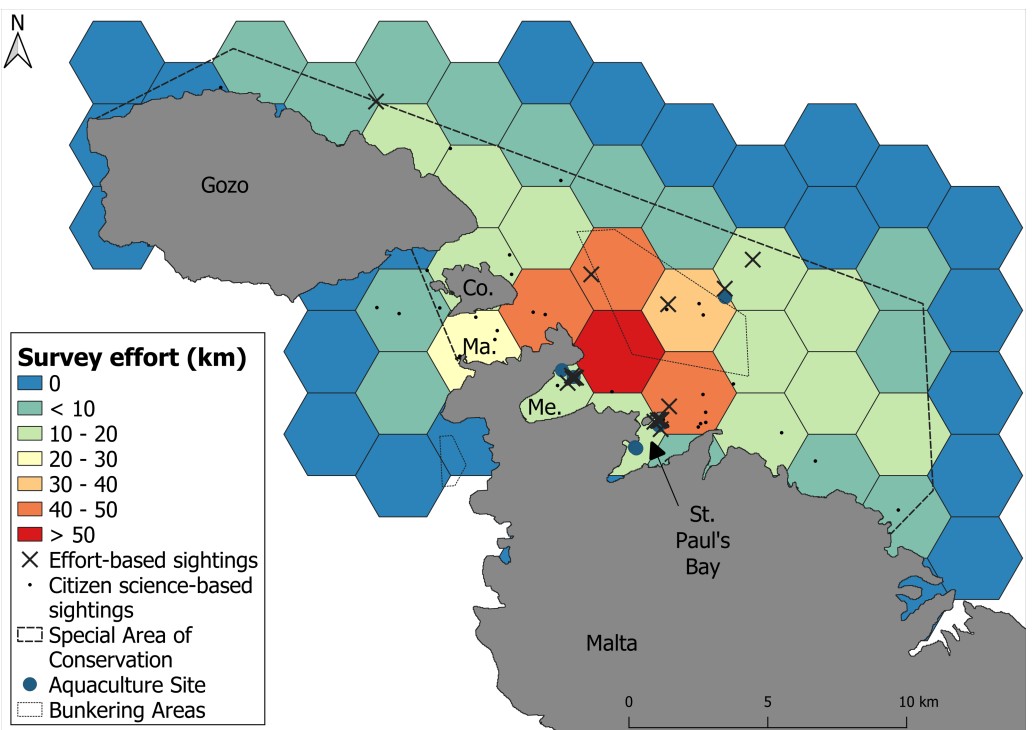

**Figure 2** Survey effort calculated as the amount of on-effort track line across the survey area. Warmer colours represent areas of higher survey effort and cooler colours represent areas of lower survey effort. Sightings of dolphins are delineated by black crosses.

**Table 2** Relative importance of explanatory variables in the generalised additive model (GAM). The sum of the proportion of the total deviance explained exceeds 1 because the method penalises more complex models.

| Variable | Deviance explained | Proportion of total deviance explained[*] |
|---|---|---|
| Chlorophyll-a | 31.4% | 0.79 |
| Distance to aquaculture | 26% | 0.65 |
| Slope | 7.89% | 0.20 |
| SST anomaly | 6.36% | 0.16 |

Notes.

*The sum of the proportion of the total deviance explained exceeds 1 because the method penalises more complex models.

gain was 1.23, compared to an unregularized gain of 1.18, indicating that the application of regularization successfully reduced overfitting to a small extent.

Chlorophyll-a was the most influential predictor in the model, contributing 61.2% of the total model gain and 25.8% of the permutation importance (Table 3). Distance to aquaculture was the second most influential variable, contributing 20.1% to model gain and 53.3% to permutation importance.

The model showed a marked decrease in habitat suitability with increasing distance from aquaculture facilities, particularly within the first 5,000 m. Habitat suitability increased

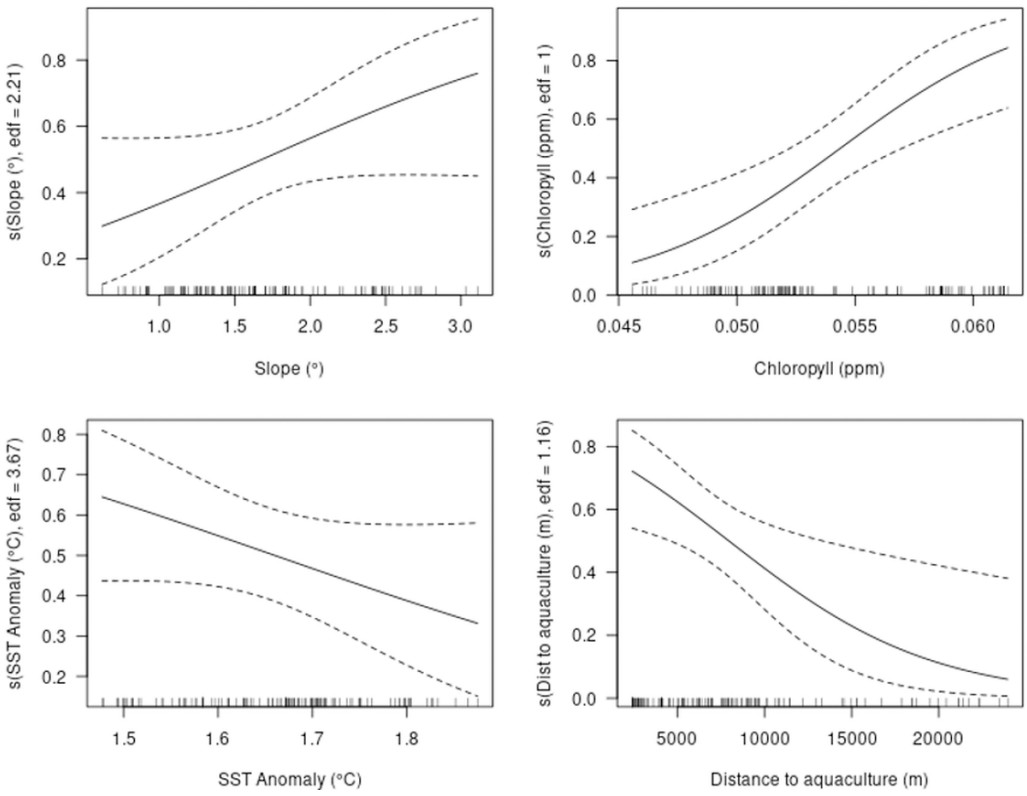

**Figure 3  Partial plots of explanatory variables for the GAM.** Slope, chlorophyll-a, sea surface temperature (SST) anomaly and distance to aquaculture.

exponentially with higher chlorophyll-a concentrations (Fig. 5). Slope and SST anomaly contributed less than 15% cumulatively to the model's predictive power. However, slope showed a higher permutation importance (20.4%), suggesting a moderate influence on model sensitivity. The response curve indicated that suitability peaked at slope values between 2.5° and 3°, declining on either side. Sea surface temperature anomaly exhibited a nearly linear negative relationship with suitability, but its overall contribution to the model was minimal (Fig. 5).

As with the GAM, the most suitable areas for bottlenose dolphins were the waters surrounding Mellieha Bay and Saint Paul's Bay. The MaxEnt model predicted high suitability only along the Maltese coast of the Comino Channel and a moderate suitability in the shallow bunkering area (Fig. 6). The coefficient of variation was highest near the edges of the survey area, and higher uncertainty was also observed within the SAC, especially along the central coastal waters of Gozo.

### Investigating the additional value of including citizen science data

When using the GAM with effort-based presences only ($n = 21$), model performance was notably lower than when research and citizen science data were combined ($n = 69$), with a mean AUC of 0.69 compared to 0.86, and a comparable TSS of 0.984 *versus* 0.951.

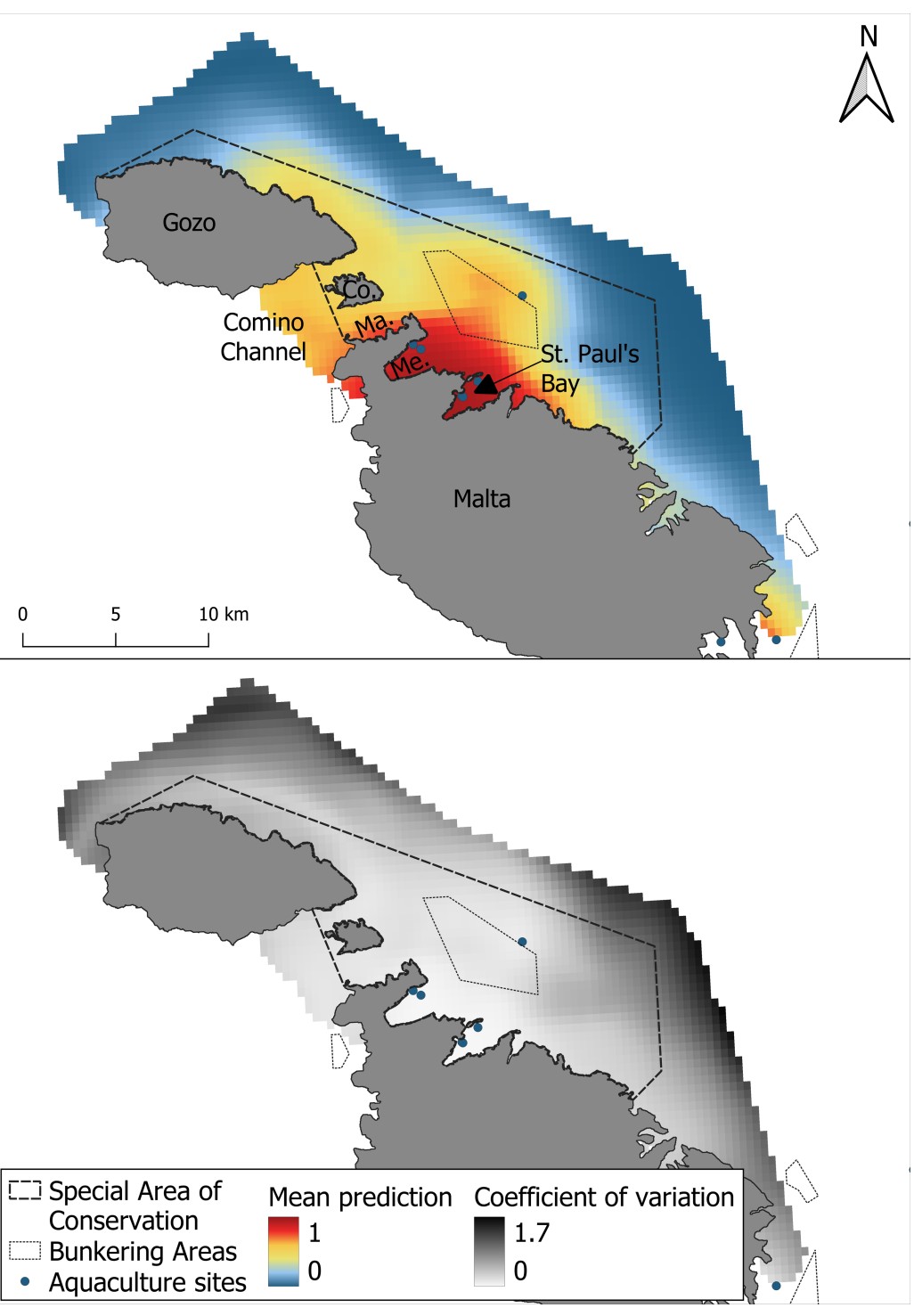

**Figure 4** Mean prediction for bottlenose dolphin habitat suitability and the coefficient of variation across 1,000 iterations of the GAM modelling process.

**Table 3  Percent contribution and permutation importance for the Maximum Entropy Model (MaxEnt).**

| Predictor | Percent contribution of predictors | Permutation importance |
|---|---|---|
| Chlorophyll-a | 61.2 | 25.8 |
| Distance to aquaculture | 20.1 | 53.3 |
| Slope | 10.4 | 20.4 |
| SST anomaly | 2.3 | 0.5 |

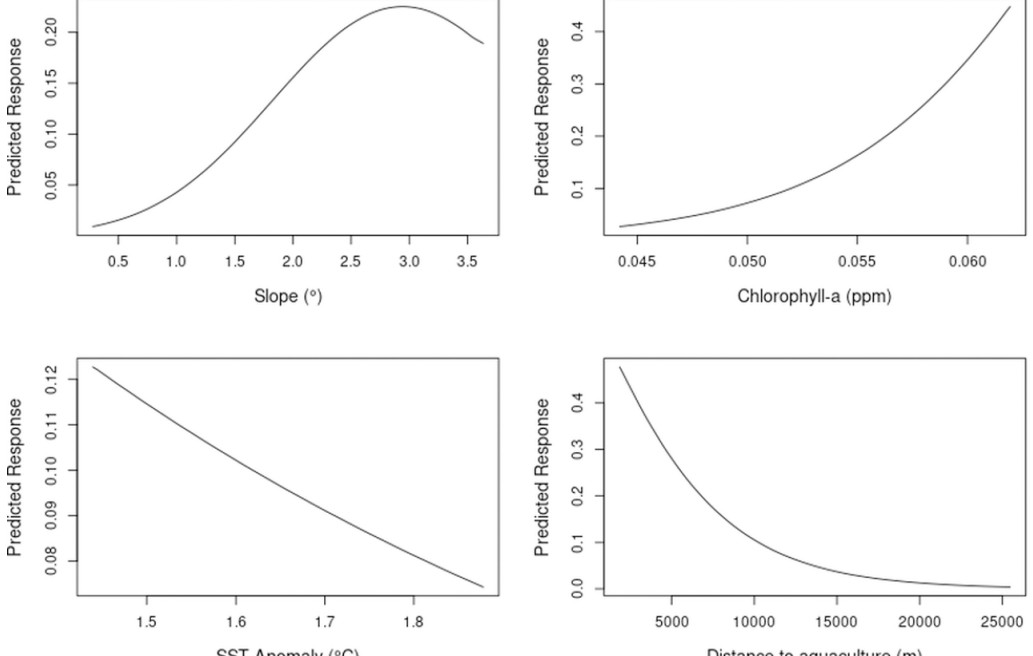

**Figure 5  Single variable response for the MaxEnt model.** Slope, chlorophyll-a, SST anomaly and distance to aquaculture.

Prediction uncertainty was also higher, with a coefficient of variation of 154.9% across the study area, particularly in the northern region (Figs. S6D–S6F). Furthermore, the limited number of data points led to an underestimation of habitat suitability in areas where dolphins were observed in the citizen science data, such as the Comino Channel (Fig. 2) and the southeast of Malta (Figs. S6A–S6C).

Although the MaxEnt model using only effort-based presences produced higher mean AUC (0.94) and TSS (0.89) compared to the model based on the full dataset (AUC = 0.89; TSS = 0.64), it showed increased prediction uncertainty, with the coefficient of variation rising by 76.5% across the survey area, particularly in the eastern region (Figs. S7D–S7F). Additionally, the model failed to predict suitable habitat in the Comino Channel and the southeastern part of the survey area (Figs. S7A–S7C).

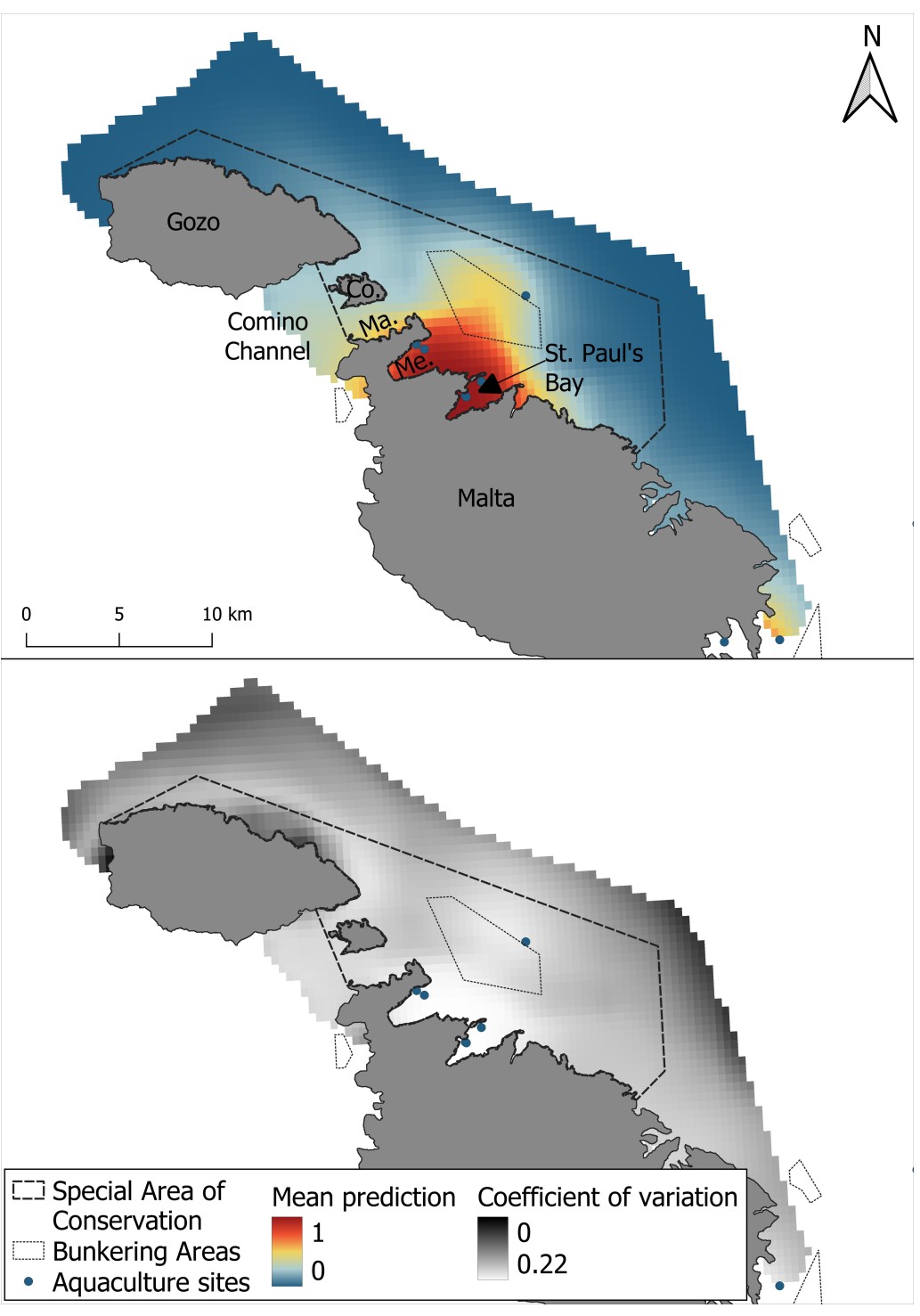

**Figure 6** Mean prediction for bottlenose dolphin habitat suitability using the MaxEnt model and the coefficient of variation across 1,000 iterations of the modelling process.

## DISCUSSION

Effective conservation planning relies on identifying areas that are important for species survival and on understanding how environmental factors influence their distribution. SDMs are widely used tools for obtaining this information, particularly in data-limited contexts (*Elith & Leathwick, 2009*; *Guisan et al., 2013*). This case study identifies core suitable habitats for a top marine predator by linking occurrence records, obtained with the support of citizen science, to key environmental variables. Through the application of a GAM and a MaxEnt model, summer habitat suitability maps were generated for the bottlenose dolphin in a relatively understudied area of Malta, offering a practical tool to support decision-makers in the formulation of conservation strategies.

This study examined the bottlenose dolphin summer habitat suitability, revealing that the northeastern coastal waters of the archipelago represent an important summer habitat for the species. In addition to being in line with previous broader research in the region (*Soster et al., 2025*), this study provides a finer-scale perspective by adding spatial detail and seasonal context to earlier habitat suitability assessments. Both the GAM and the MaxEnt models used in this research identified the area between Mellieha Bay and St Paul's Bay as a region of high suitability, suggesting that this portion of the study area may represent a key habitat for the species in summer. This area coincides with shallow, productive waters and proximity to inshore aquaculture sites, factors previously associated with higher dolphin presence in other Mediterranean regions, where similar environmental and anthropogenic features have been linked to increased dolphin presence (*Díaz López, 2012*; *Pace et al., 2019*; *Gnone et al., 2022*; *La Manna et al., 2023a*; *Bellingeri et al., 2025*). Moderate suitability was also found across the broader study area, with particular interest in the Comino Channel, but with some degree of spatial variability between the two models.

To better understand the environmental drivers behind these patterns, the relative importance of predictor variables for both models was examined. While GAM and MaxEnt differ in their analytical frameworks, both models' outputs revealed a comparable pattern in the importance of environmental variables shaping bottlenose dolphin habitat preference. Particularly, both models identified chlorophyll-a and distance to aquaculture as the most important predictors of habitat suitability, while slope and SST anomaly contributed less significantly. However, the relative contribution of these variables differed between the two approaches. In the GAM, chlorophyll-a explained the highest proportion of deviance, followed closely by distance to aquaculture. In contrast, MaxEnt revealed distance to aquaculture as the most influential predictor based on its permutation importance, with chlorophyll-a as the second most important variable. These differences reflect how each model handles variable interactions (*Guisan & Zimmermann, 2000*; *Elith et al., 2011*), but the consistency in identifying chlorophyll-a and distance to aquaculture as the top predictors highlights their ecological relevance to this population.

To further clarify these ecological patterns, the response curves offer information on how particular environmental factors affect habitat suitability. Higher chlorophyll-a concentrations showed a positive relationship with habitat suitability in both models, likely reflecting higher primary productivity and, by extension, greater prey availability (*Methion*

*et al., 2023*). Chlorophyll-a has frequently been used as a proxy for such favorable foraging conditions, given its correlation with the abundance of zooplankton and small pelagic fish (*Torres, Read & Halpin, 2008*). Indeed, its importance as a predictor of dolphin occurrence and habitat use has been demonstrated in other regions of the Mediterranean (*La Manna, Ronchetti & Sarà, 2016*; *Giannoulaki et al., 2017*; *Karamitros et al., 2020*). In this study, chlorophyll-a levels were likely influenced by both natural productivity and anthropogenic input from aquaculture (*Díaz López & Methion, 2017*), as chlorophyll-a concentrations were consistently higher in areas surrounding aquaculture sites during the summer months (Fig. S1), contributing to their predictive strength in both models. Additionally, bottlenose dolphins are highly opportunistic predators with a well-documented behavioral plasticity (*Reynolds, Wells & Eide, 2013*), often concentrating their foraging activity in areas where prey is predictably abundant. Suitability also declined with increasing distance from aquaculture sites, suggesting that these areas may provide foraging opportunities by promoting local prey aggregations (*Díaz López, 2012*; *Díaz López, 2017*; *Díaz López & Methion, 2017*). However, this does not imply that the area would be unsuitable in the absence of aquaculture; rather, the presence of these sites appears to increase habitat use within an ecologically favorable region. Finally, while the observed negative correlation between bottlenose dolphin presence and distance to aquaculture facilities improves our ability to predict their habitat preferences, this aggregation near fish farms may pose conservation concerns, as previous studies have associated such clustering with increased risks of vessel collisions, entanglement, habitat degradation, and exposure to elevated noise and light pollution (*Guisan & Zimmermann, 2000*; *Martino et al., 2021*).

While chlorophyll-a and distance to aquaculture were the strongest predictors in both models, slope and SST anomaly also contributed to explaining dolphin habitat suitability, albeit to a lesser extent. In the GAM, slope showed a significant but weaker relationship, with habitat suitability peaking at intermediate values. Similarly, the MaxEnt response curve indicated that suitability decreased on either side of an optimal slope range, suggesting a preference for specific seabed structures. Both models indicated a preference for areas characterized by gentle slopes (~2.5–3°), indicating that bottlenose dolphins in the study area are associated with mildly inclined rather than very flat or steep seabed. This is consistent with previous findings suggesting that moderate bottom inclinations may support preferred foraging or transit zones, particularly in coastal environments where bathymetric features influence prey availability (*Cañadas, Sagarminaga & Garcı'a-Tiscar, 2002*; *Gnone et al., 2022*).

Although SST anomaly was the weakest predictor in both models, the response curves indicated a consistent negative relationship between increasing temperature anomalies and habitat suitability. This suggests that bottlenose dolphins may tend to avoid areas experiencing greater thermal deviation, potentially due to indirect effects on prey availability or ecosystem stability (*Wernberg et al., 2013*). In the Mediterranean, SST anomalies have been associated with shifts in the abundance and distribution of demersal and pelagic fish species (*Sabatés et al., 2006*; *Hidalgo et al., 2011*), which could in turn reduce the attractiveness of the areas affected to predators. Given that the study only covered summer months, the role of thermal variability might be underestimated, and further investigation
across seasons would be needed to better understand its influence. This is of particular importance, in light of the potential consequences of extreme weather events such as marine heatwaves that are becoming more frequent in the Mediterranean (*Darmaraki et al., 2019*; *Wild et al., 2019*).

One important finding of this study is the spatial agreement shown by the two models in predicting areas of high habitat suitability, particularly in the waters surrounding Mellieha Bay and St Paul's Bay. However, a certain difference between the two modeling approaches was observed. Particularly, the GAM predicted a broader extent of high suitability, including the Comino Channel, as well as moderate suitability across most of the study area. In contrast, the MaxEnt model identified a more spatially restricted area of high suitability in the channel, primarily concentrated in Marfa Bay, with moderate suitability extending across the bunkering area. Additionally, while the coefficient of variation was higher around the SAC boundaries for both models, the MaxEnt predictions exhibited greater uncertainty within the SAC itself, particularly around the northern coastal waters of Gozo relative to the overall coefficient of variation. These differences likely reflect the methodological dissimilarities between the models and their respective sensitivities to the spatial distribution and structure of the input data. GAM's use of pseudo-absence data and smooth functions may have allowed it to detect localized patterns of suitability, particularly if species-environment relationships in this area are non-linear or influenced by smaller gradients (*Guisan, Edwards & Hastie, 2002*; *Wood, 2017*). In contrast, MaxEnt's reliance on background sampling and global feature fitting may have reduced sensitivity to weak or spatially clustered signals in regions with lower observation density (*Elith et al., 2011*; *Merow, Smith & Silander, 2013*). Interestingly, while MaxEnt's outputs have been found to be only similar to those of GAMs when background points are drawn from observed absences (*Fiedler et al., 2018*), these results show spatial agreement despite the use of pseudo-absences and background data, suggesting a robust ecological indicator captured by both modeling approaches. These model-specific differences highlight the value of employing complementary modeling techniques, as they can reveal different aspects of species-habitat relationships and strengthen the ecological interpretation of suitability maps (*Guisan & Zimmermann, 2000*; *Elith & Graham, 2009*).

Given its significance in conservation applications of SDMs (*Guisan et al., 2013*), it is important to highlight the low degree of uncertainty found in the central core area identified between Mellieha Bay and St. Paul's Bay. While some uncertainty is inherently irreducible (*Regan et al., 2005*), the low uncertainty and strong spatial agreement of highly suitable areas, together with the high predictive performance of both models (AUC > 0.85), reinforce confidence in the identified ecological patterns. Making the uncertainty explicit also helps stakeholders understand the level of confidence they can place in the prediction (*Regan et al., 2005*). The highly suitable area found by both models should therefore be treated as a conservation priority, while the broader GAM-predicted moderately suitable areas may inform future monitoring efforts. In data-limited contexts, this complementarity can provide a more holistic and robust foundation for decision-making.

Incorporating citizen science data was a key aspect of this study, enabling a nearly threefold increase in presence records as well as expanding the number of sightings in areas

with less survey coverage. In contrast, models based solely on research data predicted low habitat suitability in the Comino Channel, despite frequent dolphin sightings reported by citizen scientists (Fig. 2). The reduced coefficient of variation in both the GAM and MaxEnt models using the full dataset further underscored the value of including opportunistic observations, especially when systematic survey effort is limited. These findings align with previous research showing that appropriately incorporating citizen science and opportunistic data can significantly enhance species distribution models (*Tiago, Pereira & Capinha, 2017*; *Coxen et al., 2017*; *Matutini et al., 2021*). Our results contribute to the growing recognition of non-traditional data sources as valuable tools for contributing to cost-effective long-term monitoring and the generation of reliable habitat suitability predictions. These predictions help govern conservation, particularly in marine environments where systematic data collection is often constrained (*Giovos et al., 2016*; *Pace et al., 2019*).

Protecting bottlenose dolphins in Maltese waters effectively calls for the inclusion of species–habitat knowledge into site-based and pressure-based management approaches (*Reisinger, Johnson & Friedlaender, 2022*). This study adopts a conservation-oriented perspective and identifies key summer habitats which, although located within an existing SAC, are not currently designated for the protection of bottlenose dolphins. The strong agreement between habitat suitability maps highlights consistent ecological patterns and supports spatial planning, local mitigation, and the potential inclusion of bottlenose dolphins in the site's conservation objectives. Furthermore, the integration of citizen science data demonstrates how participatory monitoring expands spatial coverage and enhances model reliability in under-surveyed areas. Employing complementary modeling approaches, such as MaxEnt and GAMs, further strengthens the ecological interpretation of habitat use and provides evidence for adaptive management in data-limited marine environments.

## Study limitations

Despite providing valuable insights into the summer habitat suitability of bottlenose dolphins in Maltese waters, this work is subject to certain limitations that should be acknowledged. First, the study was limited to a single season, potentially overlooking important temporal dynamics and habitat preferences that vary across seasons. Moreover, data were collected during a single summer period, which may not fully capture interannual variability in dolphin occurrence and habitat use. As such, conclusions drawn are primarily reflective of summer habitat suitability patterns and may not represent consistent seasonal patterns across years or year-round habitat use. Secondly, although sampling bias was addressed in both modeling exercises, the integration of citizen science and opportunistic data may remain susceptible to uneven effort and reporting. These limitations may result in underrepresentation of certain areas, where lower predicted suitability could reflect limited observation effort rather than true absence. Third, the moderate spatial disagreement between the GAM and MaxEnt predictions, particularly in areas such as the Comino Channel, suggests that methodological sensitivity to data structure may affect spatial outputs. While both models showed strong AUC values, the reliance on pseudo-absence or

background data remains a known limitation in interpreting habitat suitability, particularly in data-poor or spatially heterogeneous systems (*Lobo, Jiménez-Valverde & Real, 2008*; *Phillips et al., 2009*).

## CONCLUSIONS

This study applied a comparative modeling approach using generalized additive models and MaxEnt to assess summer habitat suitability for bottlenose dolphins in a coastal Special Area of Conservation (SAC) in Malta. The models integrated citizen science and opportunistic data with environmental and anthropogenic variables to identify key drivers of dolphin habitat suitability and highlight important habitats within a data-limited context.

Both modeling approaches consistently identified shallow, nearshore regions, particularly around Mellieha Bay and St Paul's Bay, as key summer habitats for bottlenose dolphins. Chlorophyll-a concentration and proximity to aquaculture sites emerged as the strongest predictors, while slope and sea surface temperature anomaly played a secondary role. Despite some methodological differences between the models, the observed spatial agreement strengthens confidence in the ecological relevance of these findings.

By combining traditional ecological knowledge with citizen science contributions, this work demonstrates a scalable and cost-effective approach for habitat assessment, especially valuable in regions where systematic monitoring is limited. The results contribute to filling critical knowledge gaps on dolphin spatial ecology in Malta's coastal waters and offer science-based support for potential updates to conservation planning frameworks, including the consideration of bottlenose dolphins as a qualifying species within the SAC.

However, this study is not without limitations. The research focused on a single summer season and was based on presence-only data, limiting its ability to capture interannual variability and broader seasonal dynamics. Additionally, while sampling bias was addressed, the use of opportunistic data remains susceptible to uneven spatial effort and reporting.

Future research should prioritize increasing the temporal resolution of data collection across multiple seasons and years, improving systematic survey coverage, and further investigating the ecological linkages between aquaculture, productivity hotspots, and dolphin habitat use. Exploring these dynamics under different climatic scenarios, including marine heatwaves, may also provide critical insights for adaptive conservation management.

In summary, this case study contributes to the growing body of evidence supporting the integration of SDMs and participatory data collection into marine conservation planning. The approach presented here offers a valuable tool for informing spatial management decisions aimed at balancing species protection with human activities.

## ACKNOWLEDGEMENTS

This study was carried out as part of the project "Coastal Opportunities for Climate Change Adaptation through Sustainable Tourism, Research and Integrated Marine Ecosystem Management" (COASTWISE). Visual surveys were conducted during marine life observation tours with Discover the Blu. The staff of the Malta College of Arts, Science & Technology (MCAST) is gratefully acknowledged for their support and guidance during

this project. In particular, Professor Aldo Drago for providing oceanographic datasets through the STREAM platform (https://app.stream-srf.com/auth/welcome). The authors extend their gratitude to the research assistants Leah Camilleri and Joseph Cassar and to all the volunteers, participants and citizens who took part in the data collection, particularly Benjamin Metzger, Lucien Cambie, Sam Bard, Marie Delattre, Gabriela Górska and Justine Previ. The authors are especially grateful to Henry Copperstone and Daniele Baraggioli for providing observation platforms that complemented vessel-based surveys and protocol development. Finally, the authors are grateful to Martina Cutajar and Shona Brincat from the Ċirkewwa Marine Park for their collaboration in facilitating the citizen science campaign.

### Funding

This research was funded by Malta's Ministry for Education, Sport, Youth, Research and Innovation (MEYR). The funders had no role in study design, data collection and analysis, decision to publish, or preparation of the manuscript.

### Grant Disclosures

The following grant information was disclosed by the authors:
Malta's Ministry for Education, Sport, Youth, Research and Innovation (MEYR).

### Competing Interests

Timothy Zammit is the owner of Discover the Blu. Nina Vérité-Taulet is employed by Discover the Blu.

### Author Contributions

- Francesca Soster conceived and designed the experiments, performed the experiments, analyzed the data, prepared figures and/or tables, authored or reviewed drafts of the article, and approved the final draft.
- Tim Awbery conceived and designed the experiments, analyzed the data, prepared figures and/or tables, authored or reviewed drafts of the article, and approved the final draft.
- Nina Vérité–Taulet performed the experiments, authored or reviewed drafts of the article, and approved the final draft.
- Timothy Zammit performed the experiments, authored or reviewed drafts of the article, and approved the final draft.
- Kimberly Terribile conceived and designed the experiments, authored or reviewed drafts of the article, and approved the final draft.

### Animal Ethics

The following information was supplied relating to ethical approvals (*i.e.*, approving body and any reference numbers):

The animal study protocol was approved by Malta's Environment and Resources Authority (ERA), Malta (permit number EP 0249/24, issued on 12 July 2024).

## Data Availability

The bottlenose dolphin sighting data are available in the Supplementary Files.

## Supplemental Information

Supplemental information for this article can be found online at http://dx.doi.org/10.7717/peerj.19804#supplemental-information.

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
