# Peer review of "Harnessing citizen science for marine conservation in Malta: a comparative analysis of GAM and MaxEnt models in bottlenose dolphin habitat mapping"

_PeerJ, doi:10.7717/peerj.19804_

## Round 0.1 · original submission · Major Revisions

The two reviewers have provided very detailed comments in their summaries, as well as on the manuscript itself. I've no doubt that you will be able to address these quickly and do so in a way that will be compelling. Please deal with each comment in turn and be explicit in your cover letter in how you have answered them.

·

Basic reporting

The article is well written but would benefit from a review of intext referencing. Some intext references are in different formats e.g., Guisan et al. 2013 and Marshall, Glegg & Howell, 2014.
Please review the methods section of the article - there are large passages of text with no references at all (for example lines 227-261).

Please review the references to Supplementary Material in the manuscript - Supplementary Material 3 is referenced in text but only 2 Supplementary Material documents have been provided. Lines 156/157 imply that the standardised observation protocol during whale watch activities has been provided in Supplementary Material 1 but this information is missing from the Supplementary Material provided for review. This information would be useful for the reader so should be provided either in the revised Supplementary Material or summarised in the methods section.

I have made a few suggestions on the manuscript where further context or information would benefit the reader, enhancing overall clarity and impact.

Experimental design

No comment

Validity of the findings

No comment

Additional comments

Congratulations to the authors, the manuscript provides important ecological and conservation information about the bottlenose dolphin in Maltese waters. It is great to see the inclusion of citizen science data and further evidence for data of this nature's potential to support conservation.

·

Basic reporting

The authors present an interesting study on the use of citizen science and different algorithms to understand Tursiops truncatus habitat use in Malta, with potential implications for local conservation. The manuscript is well-written, clear, and concise, but some points require attention. My primary concern is that, in my assessment, the manuscript fails to address its main purpose: Is it a methodological paper, comparing algorithms and delving deeply into these technical aspects, or a conservation-oriented article where the different methods are merely a secondary detail? This distinction is crucial for balancing relevant information and determining the appropriate level of detail, particularly in the discussion. For example, a MPA manager might find it confusing to encounter varying habitat-use predictions for Tursiops truncatus depending on the algorithm applied.

Other minor—yet equally important—points include a need for conceptual clarification regarding the terms used by the authors in relation to species distribution modeling and habitat suitability. Detailed review comments are provided in the attached PDF.

Experimental design

The experimental design is clear and analysis are well described. However, I would like the authors to clarify about the validation of the citizen science data.

Validity of the findings

Data is robust and the findings are interesting. I feel that the citizen science data in the models could be better utilized by demonstrating how its inclusion improved (or did not improve) model performance.

---

## Round 0.2 · accepted · Accept

Thanks for the prompt response to the reviewers. I am now happy with this!

·

Basic reporting

No comment

Experimental design

No comment

Validity of the findings

No comment

Additional comments

No comment